# Simultaneous Probing of Metabolism and Oxygenation of Tumors In Vivo Using FLIM of NAD(P)H and PLIM of a New Polymeric Ir(III) Oxygen Sensor

**DOI:** 10.3390/ijms231810263

**Published:** 2022-09-06

**Authors:** Yulia P. Parshina, Anastasia D. Komarova, Leonid N. Bochkarev, Tatyana A. Kovylina, Anton A. Plekhanov, Larisa G. Klapshina, Aleksey N. Konev, Artem M. Mozherov, Ilya D. Shchechkin, Marina A. Sirotkina, Vladislav I. Shcheslavskiy, Marina V. Shirmanova

**Affiliations:** 1G. A. Razuvaev Institute of Organometallic Chemistry, Russian Academy of Sciences, Tropinina, 49, 603950 Nizhny Novgorod, Russia; 2Institute of Experimental Oncology and Biomedical Technologies, Privolzhsky Research Medical University, Minin and Pozharsky Sq. 10/1, 603005 Nizhny Novgorod, Russia; 3Institute of Biology and Biomedicine, Lobachevsky State University of Nizhny Novgorod, 23 Gagarin Avenue, 603950 Nizhny Novgorod, Russia; 4Becker & Hickl GmbH, Nunsdorfer Ring 7-9, 12277 Berlin, Germany

**Keywords:** phosphorescent polymeric iridium(III) complexes, bioimaging, phosphorescence lifetime imaging, oxygen sensing, tumor, in vitro, in vivo

## Abstract

Tumor cells are well adapted to grow in conditions of variable oxygen supply and hypoxia by switching between different metabolic pathways. However, the regulatory effect of oxygen on metabolism and its contribution to the metabolic heterogeneity of tumors have not been fully explored. In this study, we develop a methodology for the simultaneous analysis of cellular metabolic status, using the fluorescence lifetime imaging microscopy (FLIM) of metabolic cofactor NAD(P)H, and oxygen level, using the phosphorescence lifetime imaging (PLIM) of a new polymeric Ir(III)-based sensor (PIr3) in tumors in vivo. The sensor, derived from a polynorbornene and cyclometalated iridium(III) complex, exhibits the oxygen-dependent quenching of phosphorescence with a 40% longer lifetime in degassed compared to aerated solutions. In vitro, hypoxia resulted in a correlative increase in PIr3 phosphorescence lifetime and free (glycolytic) NAD(P)H fraction in cells. In vivo, mouse tumors demonstrated a high degree of cellular-level heterogeneity of both metabolic and oxygen states, and a lower dependence of metabolism on oxygen than cells in vitro. The small tumors were hypoxic, while the advanced tumors contained areas of normoxia and hypoxia, which was consistent with the pimonidazole assay and angiographic imaging. Dual FLIM/PLIM metabolic/oxygen imaging will be valuable in preclinical investigations into the effects of hypoxia on metabolic aspects of tumor progression and treatment response.

## 1. Introduction

In aerobic organisms, most of the oxygen supplied to the cells is consumed in mitochondrial respiration to generate energy in the form of ATP. At the cellular level, the rate of oxygen consumption is modulated by metabolic conditions, i.e., substrate availability and energetic demands. The rapid, uncontrolled proliferation of cancer cells in the absence of functional vasculature results in the development of hypoxic regions in tumors. Hypoxia plays an important role in the pathogenesis of cancer. Typically, a solid tumor contains areas of mild and severe hypoxia and, during growth, undergoes dynamic fluctuation in blood flow and, consequently, oxygen availability [1]. Hypoxia leads to multiple epigenetic and genetic alterations mediated mainly by the hypoxia-inducible factor (HIF) family that enhance cancer cell growth and aggressiveness [2,3]. Hypoxia-induced alterations in cellular metabolism depend on its severity and duration of oxygen depletion, and vary in cancers of different types and genetic background. The heterogeneous distribution of oxygen and nutrients within a tumor is supposed to be the primary cause of its metabolic heterogeneity [4]. At the same time, the most appreciated metabolic feature of tumors is their reliance on glycolysis, not only in hypoxia, but also when oxygen is available (the so-called “Warburg effect”), which means that they are generally less dependent on oxygen supply. Initially, it was postulated that the Warburg effect is due to mitochondrial dysfunction; however, it is now thought that most tumor cells preserve mitochondrial function and, in normoxic conditions, may use oxidative phosphorylation as intensively as normal cells [5]. While aerobic glycolysis is a preferable metabolic program of actively proliferating tumor cells (e.g., early in oncogenesis), tumor cells are not constitutively glycolytic and are capable of modulating their metabolism under the influence of the microenvironment, signaling events or external interventions. Overall, the relationships between oxygenation and the metabolic state of tumor cells are far from trivial and still not entirely clear.

Recently, an approach has been developed on the basis of two-photon excited laser scanning microscopy combined with the time-correlated single-photon counting (TSCPC) technique that enables the simultaneous visualization of oxygenation and metabolism at the cellular level [6]. In this approach, the assessment of oxygen content is based on the use of oxygen-sensitive phosphorescent probes administered into the cells or tissues of interest and the measurement of the distribution of probes’ phosphorescence lifetime across a sample (phosphorescence lifetime imaging, or PLIM). Due to the dynamic quenching of phosphorescence emission by molecular oxygen, phosphorescence intensity and the lifetime of the probe decrease linearly with increasing partial oxygen pressure. Imaging of the cellular metabolism is based on recording the autofluorescence of metabolic cofactors. Of the reduced nicotinamide adenine dinucleotide (phosphate) NAD(P)H and the oxidized flavin adenine FAD, NAD(P)H has a clearer interpretation and is therefore preferable. The fluorescence lifetime of NAD(P)H depends on its binding to proteins: the free form of the cofactor has a short lifetime (~0.4 ns) and is attributed to glycolysis, while the protein-bound form has a long lifetime (~2–4 ns) depending on the protein, and is attributed to the mitochondrial respiratory chain. Any changes in the activity of these pathways result in changes in the relative fractions of free and bound NAD(P)H, which can be measured using fluorescence lifetime imaging (FLIM) [7,8]. Simultaneous FLIM and PLIM is based on the modulation of a ps and fs laser pulse train, and recording fluorescence and phosphorescence decays when the laser is on and off, respectively [9]. The implementation of a simultaneous PLIM of oxygen and metabolic FLIM has been demonstrated in a few works on cultured cells and ex vivo systems [6,10]. The extension of this approach for in vivo studies is possible, but currently limited to spectroscopic point measurements with a fiber probe [11].

Over the past two decades, phosphorescent transition metal complexes have attracted great attention as oxygen sensors in living cells and tissues [12]. Among them, cyclometalated iridium(III) compounds have been found to be the most promising ones. Iridium(III) phosphores have demonstrated photophysical properties that meet the main requirements for oxygen sensing in biological objects: moderate phosphorescence lifetimes to compromise between sensitivity and data acquisition time (1–15 µs), easily tunable emission wavelengths, and relatively high quantum yields (>0.1). Iridium oxygen sensors have been developed in the form of small molecular probes and polymeric probes containing phosphorescent iridium(III) complexes comprising aromatic cyclometalating ligands and β-dicetonate or diimine ancillary ligands [13,14,15]. The obvious advantage of polymeric materials in comparison with their small molecular counterparts is their higher potential to accumulate in the tumor, owing to an enhanced permeability and retention (EPR) effect [16,17]. The emission color of iridium complexes depends mainly on the nature of cyclometalating ligands. Red-emitting complexes are especially valuable because red light most deeply penetrates into biological tissues. In addition, the general requirements of sensors for biomedical applications are low cytotoxicity and good water solubility and, for the task of simultaneous cellular-level oxygen and metabolic imaging, the ability to penetrate into living cells. The development of new efficient iridium oxygen sensors on the polymeric platform remains a challenging task.

This study is focused on the development of a new polymeric iridium(III)-based oxygen sensor applicable for simultaneous imaging with NAD(P)H and the in vivo investigation of oxygen in a tumor using PLIM for the new sensor, and metabolism using FLIM for NAD(P)H. Three polymeric probes are synthesized via ring-opening metathesis polymerization (ROMP) using oxanorbornene monomers with oligoether, amino acid groups, and norbornene monomers with an iridium(III) complex containing 1-phenylisoquinolinate cyclometalating ligands, picolinate, and pyridinylbenzimidazole ancillary ligands. Photophysical properties and sensitivity to oxygen are determined in solution. Then, the probes are tested for cytotoxicity, and the non-toxic one is investigated on cancer cells using laser scanning microscopy. The new oxygen sensor in combination with NAD(P)H FLIM is applied to correlate oxygen content and metabolic state in cultured cells and a mouse tumor model. For the first time, cellular heterogeneity in oxygen distribution and metabolism is demonstrated in vivo by simultaneous PLIM and FLIM.

## 2. Results

### 2.1. Synthesis of Polymeric Probes PIr1–PIr3

The synthesis and characterization of the polymeric probes PIr1–PIr3 (Figure 1) is described in detail in the Appendix A.

PIr1–PIr3 probes were isolated in high yields as colored gummy substances well soluble in THF, CH_2_C_l2_, CHC_l3_ and H_2_O, and insoluble in hexane. They were characterized by elemental analysis, IR and NMR spectroscopes and GPC analysis. Average molecular weights and molecular-weight distributions of polymeric probes were in the range of *M*_w_ = 22,500–62,600, *M*_w_/Mn = 1.31–1.59. The dynamic light scattering method revealed that in aqueous solutions at a concentration of 0.1–0.2 g/L, polymeric probes form nanoparticles with average sizes of 26 nm (PIr1), 32 nm (PIr2) and 26 nm (PIr3). It can be assumed that these particles are micelles, the shell of which consists of oligoether groups and amino acid fragments, and the core includes side chains with iridium complexes. GPC curves and particle size distributions of the PIr1–PIr3 probes are presented in the Appendix A.

### 2.2. Photophysical Properties of the PIr1–PIr3 Probes in Solutions

The UV/Vis spectra of the PIr1–PIr3 probes in methylene chloride and water (Figure 2, Table 1 and Table 2) contain two sets of absorption bands. The bands in the region of 260–350 nm can be assigned to spin-allowed ^1^π → π* transitions in the aromatic systems of the ligands in iridium complexes incorporated into the polymeric platform. Low-intensity bands in the region of 360–550 nm result from both spin-allowed and spin-forbidden metal-to-ligand charge-transfer transitions (MLCT) [18,19].

The wide emission bands with maxima at 588 nm (PIr1, PIr2) and 605 nm (PIr3) observed in the photoluminescence spectra of the probes in methylene chloride and water (Figure 3, Table 1 and Table 2) are associated with ^3^MLCT and ligand-centered (^3^LC) transitions in cyclometalated iridium complexes bonded to the polymeric platform [18,19,20]. The chromaticity coordinates of the photoluminescence spectra of iridium-containing probes in the CIE (Commission Internationale de l’Eclairage) diagram (Table 1 and Table 2) correspond to orange (PIr1, PIr2) and reddish-orange (PIr3) colors. The photophysical characteristics of PIr1 are close to the corresponding parameters of PIr2 (Table 1 and Table 2) and, therefore, it can be assumed that additional units with amino acid fragments in the polymer matrix of PIr2 do not noticeably change its photophysical properties in comparison with those of PIr1. A similar feature was found recently for luminescent iridium-containing polynorbornenes with cyclometalated iridium(III) complexes bonded to polymer chains via pyrazolonate anchor ligands [21].

Molecular oxygen is an active quencher of the phosphorescence of cyclometallated iridium complexes [22], and for this reason the photoluminescence intensity and quantum yields of PIr1–PIr3 in aerated solutions are appreciably lower than in degassed solutions (Figure 3, Table 1 and Table 2). Because of the higher solubility of oxygen in methylene chloride than in water [22], the quantum yields of PIr1–PIr3 in aerated methylene chloride solutions decrease significantly faster than in aerated aqueous solutions. The influence of oxygen on the photoluminescence of PIr3 in aqueous solution was investigated in more detail. It was found that the phosphorescence quantum yield decreased linearly with an increase in partial oxygen pressure (Figure 4), which is described by the Stern–Volmer equation:(1)$$\frac{\Phi_{0}}{\Phi }=1+k_{\mathrm{sv}}{\mathrm{pO}}_{2},$$
where Φ_0_ is the phosphorescence quantum yield in the deoxygenated condition; Φ is the phosphorescence quantum yield in the oxygenated conditions; k_sv_ is the Stern–Volmer constant; and pO_2_ is the partial oxygen pressure. From the curve, we found the Stern–Volmer constant to be 0.00324 mm Hg^−1^.

The lifetimes of the excited states in degassed (τ_0_) and aerated (τ, at air equilibrium) aqueous solutions of PIr1–PIr3 were determined using phosphorescence lifetime imaging (PLIM). In the degassed solutions, phosphorescence lifetimes of the probes were 3.4 μs for PIr1, 3.1 μs for PIr2, and 2 μs for PIr3 (Table 3). In aerated solutions, the phosphorescence lifetime of PIr1 decreased by 60%, and PIr2 and PIr3 decreased by 40% (Figure 5). The phosphorescence lifetimes of the probes remained stable in phosphate-buffered saline (PBS), in the presence of bovine serum albumin (BSA), and in more complex media DMEM with 10% fetal bovine serum (FBS).

### 2.3. Cytotoxicity of Polymeric Probes PIr1–PIr3

The cytotoxicity of polymer probes PIr1–PIr3 was investigated by an MTT assay (Figure 6). The half-inhibitory concentration (IC_50_) of PIr1 was 5 ± 2 µM, PIr2- 25 ± 4 µM. PIr3 showed no pronounced cytotoxicity. At a concentration of 75 µM, the percentage of viable cells was 78 ± 3% after 24 h of incubation. Therefore, PIr1 and PIr2 probes were excluded from further study due to the high cytotoxicity.

### 2.4. Cellular Uptake and Distribution of PIr3

The ability of the PIr3 polymer probe to penetrate into cancer cells in vitro was investigated using laser scanning microscopy (Figure 7). It was found that PIr3 quickly penetrates into live CT26 cells. The intense phosphorescence of the probe was observed in the cells after 15 min of incubation, which then gradually increased until 6 h (Figure 7A,B). In the time period from 30 min to 1 h, a temporal decrease in the signal intensity was detected in cells, which could be associated with the sub-cellular redistribution of PIr3 or its efflux. The analysis of the intracellular localization of PIr3 after 3 h of incubation revealed that the probe did not colocalize with the plasma membrane, lysosomes or mitochondria, likely being distributed freely in the cytosol (Figure 7D). In parallel, PLIM images of the cells were acquired. Measurements of the phosphorescence lifetimes of the PIr3 probe in the cell cytoplasm showed that the lifetime value remained stable, ~1.15 µs, throughout the whole incubation period of 9 h (Figure 7C). Due to uniform distribution of oxygen, variations in the phosphorescence lifetime between cells in the monolayer were minor.

### 2.5. Simultaneous Oxygen and Metabolic Imaging In Vitro

Using the PLIM and FLIM options on a laser scanning microscope, the phosphorescence lifetime of the PIr3 polymer probe and fluorescence lifetime of NAD(P)H were evaluated simultaneously in CT26 cells, which allowed the correlation of oxygen content and cellular metabolic state (Figure 8). Hypoxia was modeled by placing a glass coverslip over a portion of the cell monolayer and the measurements were performed at different distances from the edge of the coverslip, corresponding to different levels of cell oxygenation. With a decrease in the level of oxygenation, the phosphorescence lifetime of PIr3 increased from 1.19 ± 0.02 μs in normoxia (N, 21% O_2_) to 2.14 ± 0.08 μs in severe hypoxia (H3, ~0.1% O_2_). As expected, the relative contribution of free NAD(P)H increased in hypoxia from 82.2 ± 1.6 % (N) to 87.7 ± 1.7 % (H3), *p* = 0.0001, indicating the change in the metabolic status towards glycolysis. A strong positive correlation, *r* = 0.73, was observed between free NAD(P)H fraction and the phosphorescence lifetime of PIr3 in the individual tumor cells (Figure 8D).

### 2.6. In Vivo Study of Oxygen and Metabolism in Tumors

Next, the protocol for the in vivo use of the PIr3 probe was developed. At this stage, we compared intravenous (i.v.) vs. local intratumoral injections of the probe and selected the appropriate dose. Notably, the doses up to 40 mg/kg did not show any visible toxic effects in mice within at least 5 days of observation. It was found that, although i.v. administration resulted in the accumulation of PIr3 in the tumor, its intracellular concentration was fairly low to provide a sufficient amount of photons for building up a decay curve (photon number was 100–350 at 3 h and 350–950 at 6 h post-injection). The intratumoral injection of PIr3 provided higher photon counts (700–3000 photons) at lower doses (2–4 mg/kg) and, therefore, was more suitable. In spite of the high concentration of the probe in the subcutaneous tumor, the phosphorescence signal in tumor cells recorded through the intact skin was weak, so we had to remove the skin flap to access the tumor. In the tumor tissue, the PIr3 probe was clearly distributed both inside the cells and in the extracellular space (Figure 9A).

The developed technique of in vivo PLIM in combination with the FLIM of NAD(P)H was used to estimate the oxygen and metabolic status of eight CT26 tumors of different volumes (Figure 9). The phosphorescence lifetime of the probe recorded in vivo matched the values obtained in vitro in normoxia and during the simulation of hypoxia conditions, and ranged from 1.1 to 2.0 μs. Most of the tumors (No. 2 to 6) had a phosphorescence lifetime of ~1.7 μs, and thus were characterized by hypoxia. In two tumors (No. 7 and 8), areas of normoxia (τ PIr3 ~ 1.2 μs) were also present. The relative contribution of the free NAD(P)H a1 in the group of tumors was in the range of 75–87%. For comparison, the phosphorescence lifetime of PIr3 in the skin epithelium was 1.5 ± 0.1 µs and the contribution of free NAD(P)H was 67.4 ± 1.7% (Appendix A), which suggests that tumors are less oxygenated and have more glycolytic metabolism than normal tissues. This is consistent with previous findings [11,23].

Using Pearson’s correlation analysis, a moderate positive correlation (*r* = 0.56) was established between the phosphorescence lifetime of the polymer probe PIr3 and the relative contribution of free NAD(P)H in tumor cells (Figure 9C). This result indicates a weaker correlation between oxygen level and metabolism in the tumor tissue compared with cultured cells.

The tumors had different degrees of heterogeneity of oxygen status and metabolic status, with a tendency for the highest intercellular variability in tumors of a large size (>40 mm^3^, No. 7 and 8) (Figure 9D). The comparison between tumors of different sizes showed that the group of medium tumors (13–20 mm^3^) had a statistically longer phosphorescence lifetime of PIr3 and a greater contribution of free NAD(P)H and shorter mean NAD(P)H lifetime than small (<7 mm^3^) and large (>40 mm^3^) tumors, which testified to their more hypoxic and glycolytic states (Figure 9E, Appendix A).

A hierarchical cluster analysis on the NAD(P)H fluorescence and PIr3 phosphorescence lifetime parameters identified two clusters: a cluster containing mainly normoxic cells of large tumors (green), and a cluster containing the remaining hypoxic cells belonging to the small, medium and large tumors (red) (Figure 9F). Therefore, the obtained dendrogram supports our conclusion about the high heterogeneity of large tumors.

### 2.7. In Vivo Microvascular Network Assessment by OCT-A

In order to check if the intertumor heterogeneity in oxygen level detected by PLIM was associated with tumor vasculature, in vivo imaging of the perfused blood vessel networks was performed using OCT-A (Figure 10A). The small tumors (<7 mm^3^) had thin (≤20 μm in diameter) convoluted blood vessels. In medium-sized tumors (13–20 mm^3^), greater amounts of larger (≥40 μm in diameter) vessels were observed; however, the vascular net was less dense and contained areas lacking perfusion. Large tumors (>40 mm^3^) had a densely formed network of various caliber vessels. Quantification of the perfused vessel density (PVD) in the projection of tumor nodes showed statistically lower values for the tumors of medium size, which explains their more hypoxic status (Figure 10B).

### 2.8. Histopathological and IHC Analysis of Tumors

Histologically, CT26 tumors were characterized by a dense cellular structure with a predominance of parenchyma over stroma (Figure 11, upper row). The stroma was represented by small diffuse bundles of collagen fibers, occupying 2–3% of the neoplastic tissue, and single small capillaries, mainly of the sinusoidal type. Spontaneous necrosis was observed in each tumor and localized mainly in the center of the nodules, occupying up to 60–70% of the tumor area. To assess hypoxia, an immunohistochemical analysis was performed with a hypoxia marker pimonidazole (Figure 11, bottom row). In all tumors, the hypoxic (pimonidazole-positive) fraction was more than 60% (Figure 11B). The medium-sized tumors with a hypoxic fraction 85.6 ± 5.6% were more hypoxic than the small and large ones, which is consistent with PLIM results.

## 3. Discussion

It is known that the tumor microenvironment, including local hypoxia, has a profound effect on tumor growth, angiogenesis, metastasis, and resistance to therapy. Adaptation to changes in oxygen tension is mediated, to a large extent, by the reprogramming of metabolic pathways and changes in the levels of different metabolic intermediates. Here, we made an attempt to simultaneously track metabolic state and oxygen level in the tumors of live animals using intravital two-photon excited fluorescence and phosphorescence microscopy. For the first time, the intratumoral heterogeneity of cellular metabolism (assessed from fluorescence lifetime of NAD(P)H) was correlated to oxygen distribution (assessed from phosphorescence lifetime of oxygen probe PIr3) within a tumor.

For the assessment of oxygen level in the tumor, a new Ir(III)-based cell-permeable phosphorescent probe PIr3 was designed. Previously, small molecular probes on the basis of iridium complexes with btp, btph cyclometalated ligands, acetylacetonato and phenanthroline ancillary ligands were successfully used for the evaluation of oxygen level in living cells and tissues [13,14,24]. In order to improve water solubility and cellular uptake efficiencies, ancillary ligands in these complexes were functionalized with hydrophilic groups [13]. The incorporation of oxygen-insensitive fluorescent coumarin molecules into functionalized phenanthroline ligands allowed the use of iridium(III) complexes as ratiometric probes for sensing oxygen levels in living cells [14]. Highly oxygen-sensitive NIR-emissive Ir(III) containing polymeric probes was synthesized by the incorporation of iridium complexes with btph cyclometalated ligands as the end-group to poly(*N*-vinylpyrrolidone) (PVP) [15]. The PVP chain of the polymeric probe was functionalized then with fluorescent dyes such as Rhodamine B or NIR-797 and applied for the ratiometric measurement of oxygen levels in cancer cells and tissues in vitro and in vivo [15]. A red-emitting iridium oxygen sensor in the form of hyper-branched polymer dots was prepared on the base of the Ir(III) complex containing 1-(phenyl)isoquinolinato ligands functionalized with poly(9,9-dioctylfluorene) groups [25]. To improve water solubility and biocompatibility, the polymeric dots were covered with a poly(styrene-co-maleic anhydride) shell [25]. The synthesized iridium hyper-branched polymer probe was shown to be applicable as a sensor of oxygen levels in living cells, and simultaneously as an efficient agent for photodynamic therapy (PDT) [25]. Cyclometalated water-soluble Ir(III) complex Ir1* containing an auxiliary sulfonated diphosphine (bis(-diphenylphosphino)benzene) ligand was presented by Solomatina et al. [23]. The in vivo evaluation of Ir1* using whole-body PLIM showed its ability to sense the difference in the oxygenation of tumor and normal tissue (muscle). A recent paper by Yoshihara et al. reported on the development of green-emitting intracellular O_2_ probes based on the Ir(III) complex of PPY (tris(2-phenylpyridinato)iridium(III). The combined use of the intracellular probe PPYDM and intravascular probe BTP-PEG48 allowed the simultaneous O_2_ imaging of acinar cells and capillaries in the pancreas in mice by PLIM microscopy [26]. Some other Ir(III) complexes have also been obtained and tested on cultured cells [27]. These examples demonstrate that Ir(III) complexes are promising agents for the phosphorescent sensing of oxygen in biological samples with a high potential for in vivo applications. Meanwhile, the use of Ir(III) oxygen probes for the mapping of oxygen level in tumors is still limited.

The diversity of hypoxic conditions in a tumor makes it difficult to generalize on its effect on metabolism. The hypoxia-inducible responses in cellular bioenergetics and molecular landscape remain a subject of active research. For a better understanding of the role of oxygen in cancer cells’ metabolism, the simultaneous probing of metabolic state and oxygen level is required. PLIM oxygen sensing in combination with the FLIM of endogenous redox cofactors is a valuable instrument for this task. There are a few studies where this approach was employed for mapping the metabolic activity of cells and tissues. In particular, Jahn et al. investigated pericellular oxygen levels in isolated American cockroach salivary glands using the ruthenium-based complex Kr341 simultaneously with intracellular FAD at the microscopic scale. Upon stimulation with dopamine, a higher oxygen consumption and a decreased FAD fluorescence lifetime was detected as a result of the increased metabolic activity of salivary glands [10]. In the paper by Kalinina et al., cell metabolism and pO_2_ were observed using two-photon laser scanning microscopy in squamous carcinoma cell cultures and in a chicken chorioallantoic membrane (CAM) model of a tumor [6]. Hypoxia induced an increase in the phosphorescence lifetime of the Ru(BPY)_3_ oxygen sensor and a decrease in the fluorescence lifetime of NAD(P)H, which pointed to the change from oxidative to glycolytic phenotype during hypoxia [6]. The FLIM/PLIM of NAD(P)H and the Ru(II) complex TLD1433 correlated oxygen consumption and cellular metabolism during PDT in bladder carcinoma cells in vitro [28]. The only in vivo study is our own, in which metabolic and oxygen states of mouse colorectal tumors were explored with fiber-based luminescence lifetime spectroscopy [11]. Measurements of the fluorescence lifetime of NAD(P)H and the phosphorescence lifetime of iridium (III) complex BTPDM1 revealed a shift to a more glycolytic metabolism and reduced oxygenation in the tumor as compared to muscle. Therefore, the present research is the first time that metabolism and oxygen were assessed on a cellular level in a tumor model with a simultaneous FLIM/PLIM technique.

The heterogeneity and flexibility of metabolism between tumors, and even within distinct regions of the same tumors, has long been appreciated [4]. The microscopic FLIM of NAD(P)H has proven to be a suitable technique for the cellular-level visualization of variabilities in metabolic or redox states. Previously, using NAD(P)H FLIM, metabolic heterogeneity has been identified in different models: tumor spheroids, patient-derived organoids, mouse and patient tumor samples and chemo-treated monolayer cells [29,30,31,32]. Our results suggest that, while cellular metabolic phenotype is highly correlated (*r* = 0.73) with oxygen level in a monolayer cell culture, this association becomes less pronounced in the tumor in vivo. In a solid tumor, the metabolism of individual cells is regulated by a number of factors, besides oxygen, including some intrinsic properties of cancer cells, e.g., genetic mutations, differentiation, epithelial or mesenchymal cell state, cell cycle phases, and influence of the microenvironment, e.g., nutrient supply, extracellular matrix and interactions with stromal cells [4]. Even oxygen depletion can induce multiple alterations in cellular metabolism, depending on the severity of hypoxia and its duration [33]. Tumor models generated from standard cell lines typically lack genetic or histological cell-to-cell variability. Therefore, it can be assumed that the observed metabolic variability of tumor cells in similar oxygen conditions is mainly due to the microenvironment.

It is well-established that the level of hypoxia fluctuates during tumor development [34]. The major cause of hypoxia in tumors is a lack of functional blood vessels. Classically, the first peak of hypoxia occurs at an early stage of tumor growth (~2–3 mm^3^), when the blood supply from pre-existing vasculature is insufficient to support the growing tumor mass. Hypoxia activates the HIF pathway, which induces the expression of genes involved in angiogenesis (VEGF, PDGF, SDF) to increase the blood supply and O_2_ level. Angiogenesis transiently restores oxygen homeostasis, until the proliferating tumor cells do not outgrow their vasculature bed. The aberrant vessel structure and a high rate of oxygen consumption by metabolically active cells are the additional factors that promote hypoxia. As a result, the regions of acute and chronic hypoxia develop in most solid tumors [33]. In our study, smaller tumors (<20 mm^3^) were mainly hypoxic, as followed from PLIM and the pimonidazole assay; of those, the smallest ones (<7 mm^3^) were slightly better oxygenated, which is related to better perfusion, as the OCT-A showed. Larger tumors (>40 mm^3^) contained both regions of hypoxia and normoxia and, therefore, were more heterogeneous in oxygen distribution and metabolic states than the smaller ones.

There is no consistency between different studies on the evaluation of tumor hypoxia at various stages of tumor growth or in tumors of different size, even for tumors of one type. Much clinical data suggest that there is a tendency for an increase in hypoxic fraction with increasing primary tumor volume [35]. Animal models demonstrate contradictory data. For example, Kiraga et al., using three methods (analysis of hypoxia-related gene expression, positron emission tomography with [^18^F]fluoromisonidazole, and fluorescence microscopy with pimonidazole), showed that in CT26 tumors, hypoxia increased from day 8 to 14 of growth and then gradually decreased [36]. In our previous work, we did not observe any differences in oxygen level (assessed from PLIM) and hypoxic fraction (assessed from pimonidazole) between CT26 tumors at days 7, 10 and 14 [37]. Significant fluctuations in oxygen saturation (%sO_2_) levels of individual tumors were observed in head and neck squamous cell carcinoma xenografts as measured with photoacoustic imaging, but there was no correlation between tumor volume and %sO_2_ [38]. According to Shibamoto et al., the hypoxic fraction varies among mouse tumors of different types, sizes, and sites [39].

One of the major limitations of intravital FLIM/PLIM microscopy is the restricted imaging depth in tissue (~100 µM maximum in most tissues at two-photon excitation) and a small field of view (typically < 1 mm). Hence, only a limited area on the tumor surface can be inspected in live animals within a reasonable time. Given inherent differences in vascular morphology between the tumor center and periphery, and consequently, in the level of oxygenation, one cannot be confident that the imaged area is representative of a whole tumor. A possible solution to improve the image contrast and depth in two-photon microscopy is to use optical clearing agents, such as glycerol or polyethylene glycol; however, their applicability in the time-resolved imaging is still debatable because they can distort fluorescence lifetimes [40,41].

A limitation of this study is that phosphorescence lifetimes recorded from cultured cells and tumors cannot be directly converted into pO_2_ using a calibration curve made from solutions, because the cellular environment results in their elongation. Thus, the sensor has to be calibrated in cultured cells of interest and placed in changing oxygen concentrations. Unfortunately, our laser scanning microscope is not equipped with a multigas incubator chamber. The disturbance of phosphorescence decay parameters in the cellular or tissue microenvironment seems to be a common issue for different oxygen probes [23,27,42].

In the developed protocol of dual FLIM/PLIM imaging in vivo, the oxygen-sensitive probe PIr3 is administered into the tumor locally. Although intravenous injection has also been tested, PIr3 accumulation in the tumors was quite low, which can be a problem for collecting the appropriate number of photons for phosphorescence decay fitting. It should be mentioned that none of the commercial phosphorescent oxygen sensors (e.g., NanO2 by Luxcel Biosciences Ltd., Cork, Ireland or Oxyphor PtG4 by Oxygen Enterprises Ltd., Philadelphia, PA, USA) are suitable for the intracellular assessment of oxygen in vivo. The delivery of the probe via intravenous administration seems more physiological and less invasive than intratumoral injection, but requires the improvement of the probe’s selectivity to the tumor. Polymeric systems for the delivery of drugs and bioimaging agents are well known and widely used in biomedical research. Several biocompatible metal-containing polymers were prepared by Suzuki coupling polymerization [43,44], free-radical polymerization [15,45], and ring-opening metathesis polymerization (ROMP) [46]. The ROMP method is the most attractive synthetic route for preparing metal-containing polymers because ROMP reactions proceed as a rule in a controllable fashion and allow us to obtain polymeric materials with desired structure and composition [47,48]. Recently, we synthesized—via ROMP—water-soluble red-emitting polymers containing phosphorescent iridium complexes with 1-phenylisoquinoline cyclometalating ligands and pyrazolonate ancillary ligands [21], but unlike PIr3, those polymeric materials did not penetrate into cancer cells or normal skin fibroblasts, and could not be used for intracellular oxygen measurements. The selectivity of the PIr3 probe can potentially be improved by increasing the size of the micelles to elongate their blood circulation time, or by the active targeting of the specific ligands on cancer cell surfaces. We will continue the work on the optimization of the in vivo protocol, completing biodistribution and toxicity studies and the modification of the probe to achieve effective delivery to tumor cells upon systemic injection.

## 4. Materials and Methods

### 4.1. Synthesis of Polymeric Probes PIr1–PIr3

The synthesis and characterization of the polymeric probes PIr1–PIr3 (Figure 1) and the starting iridium-containing monomers are described in the Appendix A. Methods of the synthesis are presented in detail elsewhere [49,50,51,52,53,54,55,56].

### 4.2. Absorption and Emission Spectra

Electronic absorption spectra of PIr1–PIr3 in CH_2_Cl_2_ and H_2_O were registered using a Perkin Elmer Lambda 25 spectrometer. Photoluminescence spectra were registered using a Perkin Elmer LS 55 spectrometer. Quantum yields of photoluminescence of PIr1–PIr3 in CH_2_Cl_2_ and H_2_O were determined at room temperature: λ_ex_ 360 nm. Quantum yields for polymeric probes were calculated relative to Rhodamine B in ethanol (Φf 0.70), as described in [57].

### 4.3. Phosphorescence Lifetime Measurements

Phosphorescence lifetimes of the PIr1–PIr3 probes in solutions were measured using a confocal macro-FLIM/PLIM system (Becker & Hickl GmbH, Berlin, Germany) equipped with the hybrid detector HPM-100-40 and a single-photon counting card SPC-150 [9,58]. Solutions of the probes were placed either in Eppendorf tubes (aerated solutions, 50 µM) or in sealed glass tubes (degassed solutions, 50 µM). Polymer probes PIr1–PIr3 in a concentration of 10 µM were dissolved in PBS or in phenol red-free DMEM FluoroBrite (Thermo Fisher Scientific, Waltham, MA, USA) with or without the addition of 10% fetal bovine serum (FBS) (Gibco, Carlsbad, CA, USA) and imaged at room temperature.

Phosphorescence was excited with a picosecond diode laser at 375 nm (BDL-SMN-375, Becker & Hickl GmbH, Germany) and detected in the range of 607–682 nm (bandwidth filter HQ640/75, Chroma, Boston, MA, USA).

The PLIM data were processed using SPCImage 8.5 software (Becker & Hickl GmbH, Berlin, Germany). The phosphorescence decay curves were fitted with a single-exponential decay model with an average goodness of fit < 1.2. The average number of photons per curve was >5000. Image collection time was 120 s.

### 4.4. Cell Culturing

The murine colon carcinoma CT26 cells were cultured in DMEM (Gibco, Carlsbad, CA, USA) and supplemented with 10% fetal bovine serum (FBS) (Gibco, Carlsbad, CA, USA), 2 mM glutamine (Gibco, Carlsbad, CA, USA), 10 µg/mL penicillin (Gibco, Carlsbad, CA, USA), and 10 mg/mL streptomycin (Gibco, Carlsbad, CA, USA) in a humidified incubator at 37 °C and 5% CO_2_. The passaging of cells was carried out at a confluence of 70–80% with trypsin-EDTA (Thermo Fisher Scientific, Waltham, MA, USA).

### 4.5. MTT Assay

Cytotoxicity of the polymer probes PIr1–PIr3 was determined by the MTT assay. For the MTT test, CT26 cells were seeded on 96-well culture plates with 1 × 10^3^ cells per well in 200 µL of culture medium and incubated for 24 h. Polymer probes were added to the cells at concentrations of 0–75 µM and incubated for 24 h at 37 °C at 5% CO_2_. Then, the nutrient medium was replaced with MTT reagent 3(4,5-dimethyl-2-thiazolyl)-2,5-diphenyl-2H-tetrazole bromide (Alfa Aesar, Haverhill, MA, USA) at a concentration of 0.5 mg/mL in accordance with the manufacturer’s protocol, and cells were incubated for an additional 4 h. The formazan crystals were dissolved in 100 µL DMSO and the absorption was measured at 570 nm using a multimode microplate reader (Synergy Mx; BioTek Instruments, Winooski, VT, USA). The percentage of viable cells relative to the control was determined for each well. For each concentration of polymer probes, the MTT assay was performed in 3 repetitions by 10 wells.

### 4.6. Laser Scanning Microscopy

The experiments were performed using a laser scanning microscope LSM 880 (Carl Zeiss, Berlin, Germany) equipped with a TCSPC-based module (two hybrid detectors, HPM-100-40; two single-photon counting cards, SPC-150; Becker & Hickl GmbH, Berlin, Germany) and a Ti:Sa femtosecond laser MaiTai HP (Spectra-Physics Inc., Milpitas, CA, USA). The images were obtained using a water-immersion lens C-Apochromat 40×/1.2 NA.

For microscopic studies, CT26 cells were seeded in 35 mm glass-bottomed FluoroDishes (Ibidi GmbH, Gräfelfing, Germany) in the amount of 3 × 10^5^ cells in 2 mL of DMEM and incubated for 24 h (37 °C, 5% CO_2_). Then, the cells were washed with PBS and placed in FluoroBright DMEM (Thermo Fisher Scientific, Waltham, MA, USA) containing 10% FBS (Gibco, Carlsbad, CA, USA) and 10 µM of PIr3.

Cellular uptake was monitored for 9 h in the same dish placed in the microscope incubator (37 °C, 5% CO_2_). In one-photon mode, the phosphorescence of the polymer probe was excited at a wavelength of 405 nm using a diode laser and recorded in the range of 600–715 nm. In two-photon mode, phosphorescence was excited at a wavelength of 750 nm and detected in the range of 570–640 nm (bandwidth filter HQ605/70, Chroma, Boston, MA, USA). At each time-point, phosphorescence intensity and lifetime images were collected from 3 to 5 fields of view. PLIM image acquisition time was 120 s, which allowed collecting ~2000 photons per decay curve. The average power applied to the samples was ~6 mW.

Quantitative analysis of the phosphorescence intensity of the PIr3 in cells was carried out using ImageJ software (National Institutes of Health, Bethesda, MY, USA). The background signal taken from the area without cells was subtracted. The intensity was measured in the cytoplasm of individual cells using ROIs.

### 4.7. Subcellular Localization Assay

To assess the subcellular localization of the complex, CT26 cells were seeded in 35 mm glass-bottomed FluoroDishes (Ibidi GmbH, Gräfelfing, Germany) in the amount of 3 × 10^5^ cells and incubated overnight. Then, the cells were washed with PBS and placed in FluoroBright DMEM (Thermo Fisher Scientific, Waltham, MA, USA) containing 10% FBS (Gibco, Carlsbad, CA, USA).

CT26 cells were incubated with 10 µM of PIr3 for 30 min, then washed with PBS and stained with organelle-specific dyes. Lysosomes were stained with LysoTracker Yellow HCK-123 (Thermo Fisher Scientific, Waltham, MA, USA) at a concentration of 3 μM, and plasma membrane was stained with CellMask Green dye (Thermo Fisher Scientific, Waltham, MA, USA) at a concentration of 50 nM, according to the manufacturer’s protocol. The fluorescence of LysoTracker and CellMask Green was excited at a wavelength of 488 nm using a diode laser and recorded in the range of 550–580 nm and 500–570 nm, correspondingly. Mitochondria were identified by autofluorescence of the cofactor NAD(P)H; excitation at 750 nm; and detection in the range of 450–490 nm (bandwidth filter HQ470/40, Chroma, Boston, MA, USA). The colocalization of the PIr3 complex and organelle-specific dyes was assessed in ImageJ software (National Institutes of Health, Bethesda, MY, USA) using the Jacob plugin to determine the Mander’s coefficient M1.

### 4.8. Tumor Model

In vivo experiments were performed on female Balb/C mice weighing 20–22 g. All animal protocols were approved by the Local Ethical Committee of the Privolzhsky Research Medical University (Nizhny Novgorod, Russia).

To generate tumors, the suspension of CT26 cells—200,000 cells in 20 µL PBS—was inoculated intracutaneously into the external auricle tissue [59]. In vivo studies were performed on the 14th day of tumor growth. In total, 14 animals were used in the study (6 for optimization of the protocol and 8 for PLIM/FLIM study).

The tumor volume was measured using a caliper, and calculated using the formula v = π/6 × a × b × c [60].

For in vivo studies, mice were anesthetized by an intramuscular injection of Zoletil (40 mg/kg, Virbac SA, France) and 2% Rometar (10 mg/kg, Spofa, Czech Republic).

During the development of the in vivo PLIM protocol, several conditions were tested: intravenous injection of the PIr3 probe via tail vein in the dose of 20 mg/kg and 40 mg/kg with observation of the phosphorescence at 3 and 6 h after the injection; local injection of PIr3 into a tumor at the dose of 2–4 mg/kg depending on the tumor volume (in total 10–20 µL of 10 µM solution, 3–4 local injections) 20–30 min prior to imaging; imaging of subcutaneous tumor through the skin and with the skin flap opened. Tumor without the injection of PIr3 was used as a control. The group of 6 testing mice were under observation for 5 days and then sacrificed.

Immediately after the PLIM/FLIM experiment, 8 mice were sacrificed by an overdose of anesthesia (95% isoflurane) to avoid hemorrhage and tumors were frozen in an OCT compound (Sakura Finetek USA, Inc., Torrance, CA, USA) for cryopreservation.

### 4.9. Simultaneous PLIM and FLIM

Time-resolved imaging of fluorescence (FLIM) and phosphorescence (PLIM) was performed on a laser scanning microscope LSM 880 (Carl Zeiss, Berlin, Germany). The fluorescence of NAD(P)H and the phosphorescence of the PIr3 polymer probe were excited simultaneously in two-photon mode at a wavelength of 750 nm. Fluorescence and phosphorescence signals were separated by a beam splitter 495 LP (Chroma, Boston, MA, USA). Fluorescence was detected using a 470/40 bandpass filter, and phosphorescence was detected using a 605/70 bandpass filter. The image acquisition time was 120 s in cultured cells and 180 s in tumors. The number of photons in the pixel ranged from 500 to 3000 for PLIM and from 1500 to 3000 for FLIM. The binning option was used to adjust the number of photons to 5000.

The CT26 cells were incubated with 10 µM of polymer probe PIr3 for 3 h. The method of hypoxia modeling was adapted from Conway et al. [61]. Briefly, the cell monolayer was covered with a Menzel Gläser cover glass (Thermo Fisher Scientific, Waltham, MA, USA) (cover glass diameter—18 mm; thickness—170 ± 5 µm) and incubated for 2 h at 37 °C, 5% CO_2_, which allowed us to obtain a variable oxygen concentration from ~0.1% at the center of the glass to ~5% at the edge. PLIM and FLIM images were collected from three zones under the glass corresponding to different levels of hypoxia and from the zone outside of the glass corresponding to normoxia. During the experiment, cancer cells were maintained in the XL multi S Dark LS incubator (PeCon GmbH, Meerbusch, Germany) at 37 °C and 5% CO_2_.

Simultaneous intravital imaging of PIr3 and NAD(P)H in mouse tumors was performed by adapting previously established protocols for FLIM and PLIM in vivo [21,23,62,63]. Briefly, anesthetized mice with a tumor on the ear received an intratumoral injection of PIr3 20–30 min before PLIM. Immediately prior to tumor imaging, the skin flap above the tumor was carefully cut to expose the tumor, not damaging the vasculature. The ear was mounted on the glass coverslip (d = 40 mm) and secured with surgical tape, and the mouse was placed on a heating microscope stage at 37 °C. A quick search of the zone of interest and focusing were performed in the NAD(P)H fluorescence intensity mode (ex. 750 nm, reg. 450–500 nm). Then, the microscope was switched into TCSPC mode and PLIM and FLIM images were acquired simultaneously (Figure 12). 

FLIM and PLIM images were obtained from 5 to 10 fields of view in each tumor and further processed using SPCImage 8.5 software (Becker & Hickl GmbH, Germany). The phosphorescence decay curves were fitted with a mono-exponential decay model. The fluorescence decay curves were fitted with a bi-exponential model. The average goodness of fit X^2^ was <1.2. From the NAD(P)H decay curves, the short and long lifetimes (τ_1_ and τ_2_, ns), their relative amplitudes (a_1_ and a_2_, %), and amplitude-weighted (mean, τ_m_) lifetime were assessed.

### 4.10. Optical Coherence Tomography Angiography (OCT-A)

The microvasculature of the tumor model was assessed using a homebuilt multimodal OCT system (designed at the Institute of Applied Physics RAS, Nizhny Novgorod, Russia) with a central wavelength of 1310 nm, power output of 15 mW, lateral resolution of 25 µm, axial resolution of 15 µm, scanning depth of up to 1.5 mm, and a scanning speed of 20,000 A-scans per second. The system is capable of 3D microangiographic visualization based on the high-frequency filtration (>96 Hz) of partially overlapped OCT data with a finite impulse response filter in the signal space; the implementation of OCT-A is described in detail elsewhere [64]. On the basis of the OCT-A data, the perfusion vessel density (PVD) was calculated as the number of pixels of all vessel skeletons in the analyzed image area divided by the total number of pixels in this area [59].

### 4.11. Histopathology and Immunohistochemistry

Cryo-samples of the tumors were cut on a Leica CM1100 cryostat (Leica Microsystems, Wetzlar, Germany) into 7 µm thick slices. For each tumor, 3–4 slices were stained with hematoxylin and eosin according to the standard protocol and examined using a Leica DFC290 microscope (Leica Microsystems, Wetzlar, Germany).

The hypoxic fraction in the tumor tissue was analyzed using pimonidazole hydrochloride (Hypoxyprobe-1, Chemicon International, Temecula, CA, USA). Hypoxyprobe-1 was administered to mice intraperitoneally at a dose of 60 mg/kg 90 min prior to sacrification. Cryosections were stained with IgG1 mouse monoclonal antibodies conjugated with FITC at a dilution of 1:200 at 37 °C in a wet chamber for 3 h. The obtained tumor slices were examined on a Leica DMIL 31 LED fluorescence microscope (Leica Microsystems, Wetzlar, Germany), fluorescence was excited at 488 nm, and a signal was recorded in the range of 500–540 nm. The relative hypoxic fraction was calculated in the ImageJ software (NIH, Bethesda, MY, USA) as a percentage of the pimonidazole-positive area of the total tumor area.

### 4.12. Clustering Analysis

A hierarchical clustering was performed using Ward’s linkage test. Clustering was performed on in vivo cellular measurements of phosphorescence lifetimes of the Pir3 probe, fluorescence lifetimes of NAD(P)H τ_1_ and τ_2_, and their relative contributions a_1_ and a_2_. Hierarchical clustering is based on measuring the distance between objects. Clustering starts with each observation as a separate cluster and gradually combines them as the distance increases. The result presents a tree-like structure showing the distance between objects and, accordingly, the degree of their similarity, which allows the user to find the appropriate cluster. The default threshold 0.7 of the maximum distance was used for cluster separation.

### 4.13. Statistical Analysis

The obtained data were checked for the normal distribution using the Kolmogorov–Smirnov criterion. The data with a normal distribution are presented as mean ± standard deviation (SD). The data with a distribution other than normal are presented as a median and Q1 and Q3. A two-tailed Student’s *t*-test with Bonferroni correction or the Wilcoxon *t*-test were used to compare the data, with *p* < 0.05 being considered statistically significant. The Pearson correlation was calculated between the phosphorescence lifetime and NAD(P)H a_1_. Statistical data processing was performed in RStudio software (Boston, MA, USA).

## 5. Conclusions

Most tumors are characterized by a heterogeneous distribution of oxygen and the development of hypoxic regions of different severity. At the same time, cancer cells are capable of reprogramming their metabolism by a shift towards increasing glycolysis, even in the presence of oxygen. Both hypoxia and glycolytic metabolism support tumor aggressiveness and resistance to radio-, immuno- and chemotherapies. A method for the parallel monitoring of oxygen content and metabolism by dual PLIM/FLIM reported here will be a valuable approach to gain a deeper insight into the mechanisms of resistance and validate novel drugs targeting tumor hypoxia or metabolic pathways in preclinical studies. Applications of this technique and the new oxygen sensor PIr3 are not limited to cancer research, but can be extended to different pathologies associated with oxygen depletion and/or altered metabolism. In addition, red-color fluorophores can be imaged simultaneously with NAD(P)H and PIr3, and therefore, with the use of red sensors, the panel of the measured parameters can be expanded.

## Figures and Tables

**Figure 1 ijms-23-10263-f001:**
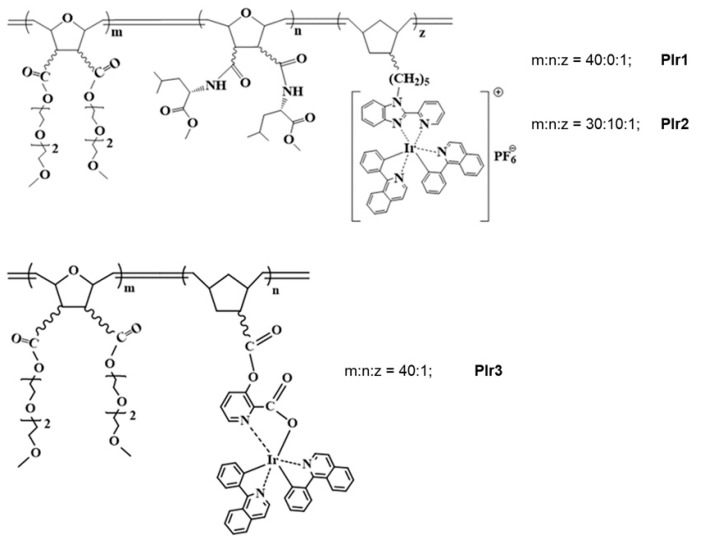
Chemical structures of the polymeric probes PIr1–PIr3.

**Figure 2 ijms-23-10263-f002:**
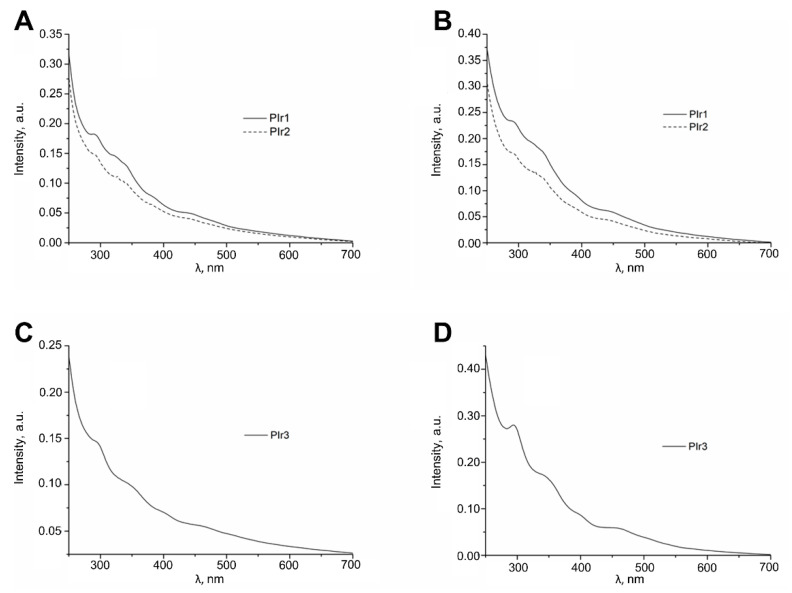
Absorption spectra of PIr1–PIr3 in CH_2_C_l2_ solution (**A**,**C**) and in H_2_O solution (**B**,**D**).

**Figure 3 ijms-23-10263-f003:**
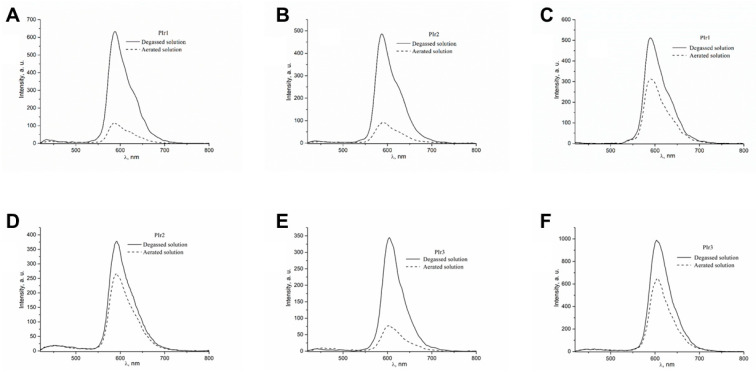
Photoluminescence spectra of PIr1–PIr3 in (**A**,**B**,**E**) CH_2_C_l2_ solution and (**C**,**D**,**F**) aqueous solution at room temperature. λ_ex_ = 360 nm.

**Figure 4 ijms-23-10263-f004:**
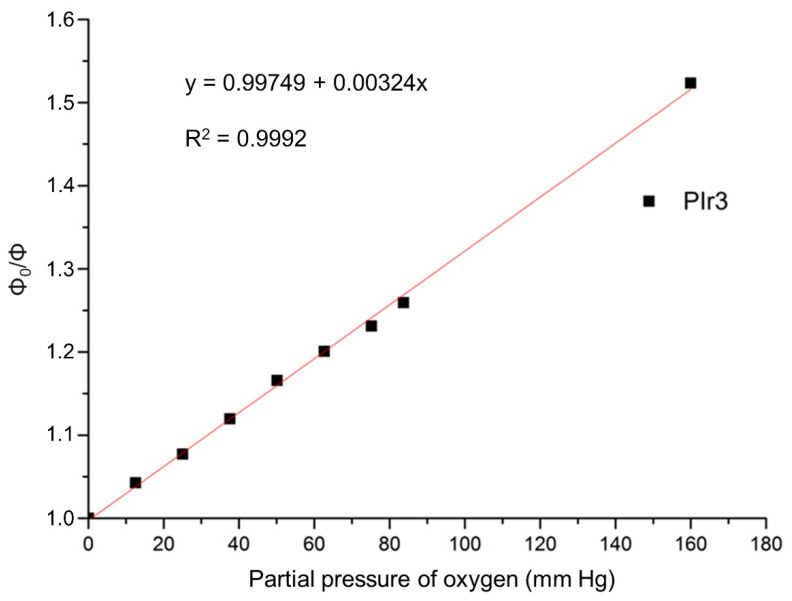
Stern–Volmer oxygen quenching plot for PIr3 in aqueous solution.

**Figure 5 ijms-23-10263-f005:**
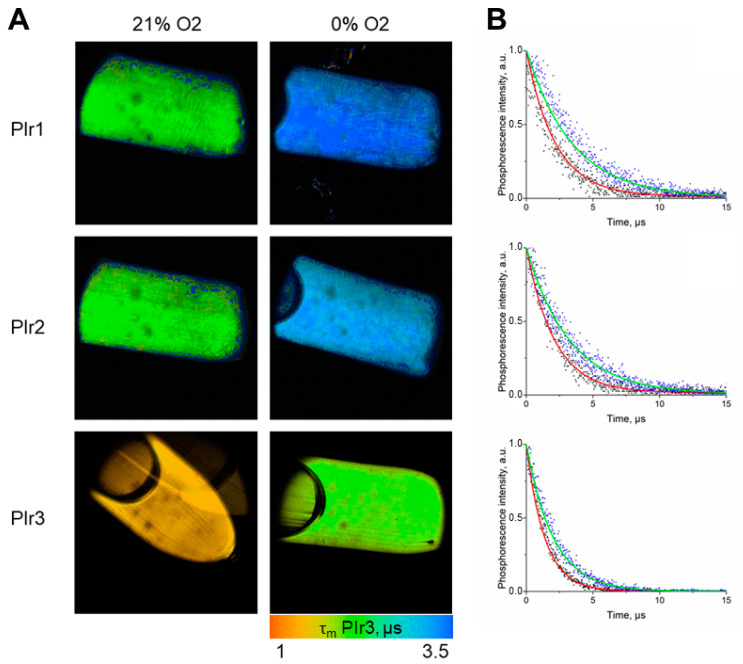
Phosphorescence lifetime measurements for PIr1, PIr2 and PIr3 in degassed and aerated aqueous solutions. (**A**) PLIM images of the tubes with solutions. Image size is 16 × 16 mm. (**B**) Phosphorescence decay curves of PIr1–PIr3 probes. On the phosphorescence decay curves, photons are indicated by dots (black—21% O_2_; blue—0% O_2_), and the lines indicate the mono-exponential phosphorescence decay curve (the red line is 21% O_2_; the green line is 0% O_2_).

**Figure 6 ijms-23-10263-f006:**
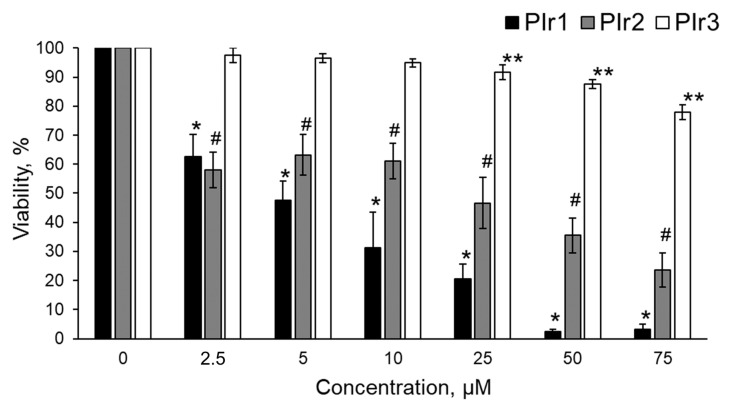
MTT assay for viability of CT26 cells after incubation with PIr1, PIr2, and PIr3 polymer probes. Mean ± SD. *, **, # *p* ≤ 0.05 vs. controls (0 µM). *n* = 3 repetitions.

**Figure 7 ijms-23-10263-f007:**
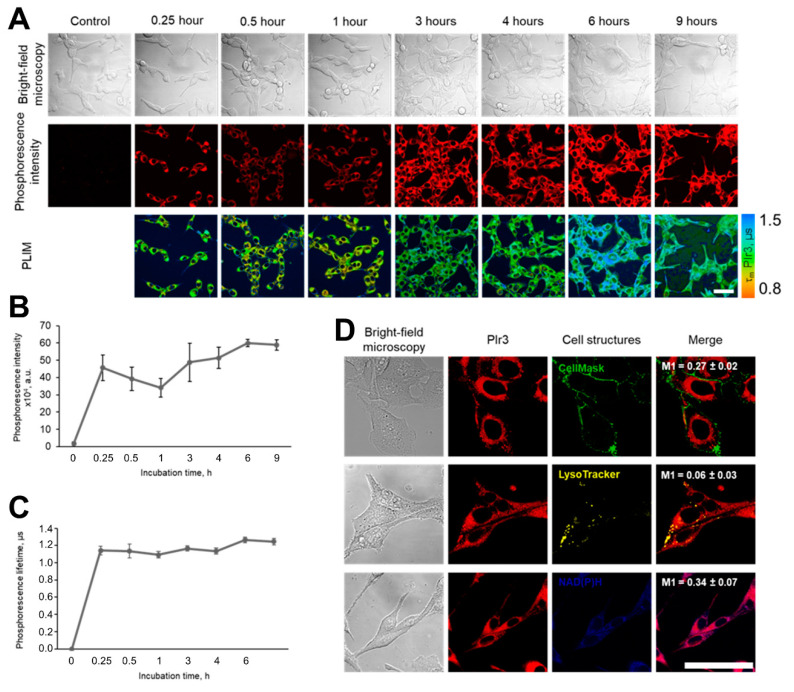
Cellular uptake and intracellular distribution of PIr3 in CT26 cells. (**A**) Bright-field, phosphorescence intensity and lifetime images obtained by laser scanning microscopy in the course of incubation with PIr3. (**B**) Phosphorescence intensity of PIr3 in cells. Mean ± SD, *n* = 70 cells. (**C**) Phosphorescence lifetime of PIr3 in cells. Mean ± SD, *n* = 50 cells; (**D**) Intracellular localization of PIr3. Phosphorescence of PIr3 is shown in red. Lysosomes were stained with LysoTracker Yellow HCK-123 (yellow), plasma membrane was stained with CellMask Green dye (green), and mitochondria were visualized by autofluorescence of NAD(P)H cofactor. Scale bar = 50 μm.

**Figure 8 ijms-23-10263-f008:**
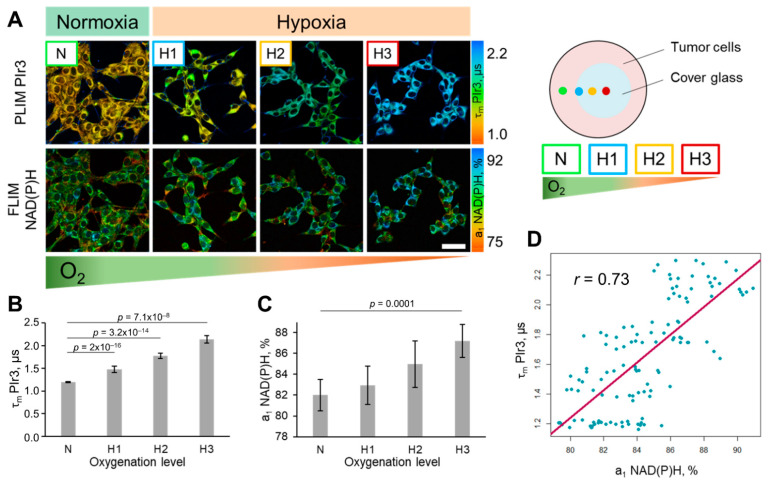
Simultaneous probing of oxygen and metabolism of CT26 cells in normoxic and hypoxic conditions. (**A**) PLIM images of cells with PIr3 and FLIM images of NAD(P)H from the same cells upon hypoxia induction. Scheme of the experiment on modeling hypoxia using cover glass. Scale bar = 50 μm. For FLIM: ex. 750 nm, reg. 450–490 nm. For PLIM: ex. 750 nm, reg. 570–640 nm. (**B**) Dependence of the phosphorescence lifetime of PIr3 in cells on the level of oxygenation. (**C**) Relative contribution of free NAD(P)H (a_1_, %) at different levels of oxygenation. Mean ± SD, *n* = 30 cells. *p*-values are indicated on the diagram. (**D**) Correlation between τ PIr3 and a_1_ NAD(P)H. The Pearson coefficient *r* is shown on the scatter plot. Blue dots are the measurements from individual cells.

**Figure 9 ijms-23-10263-f009:**
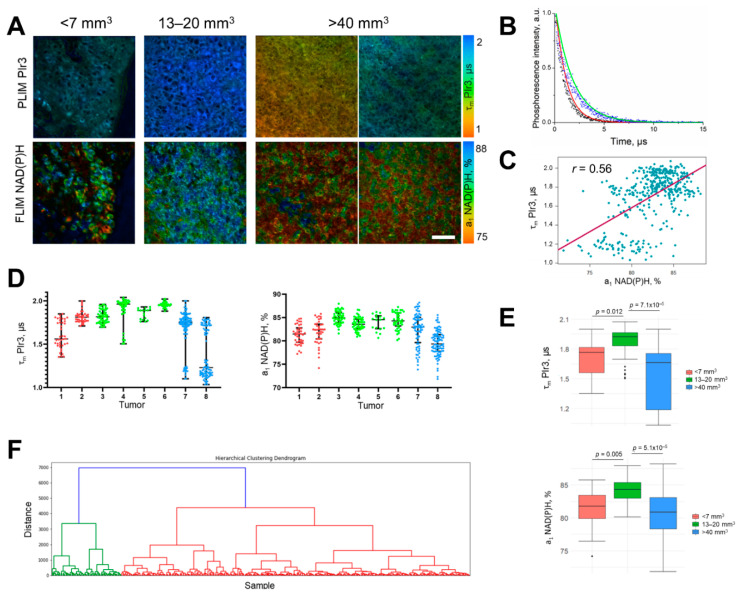
Simultaneous oxygen PLIM and metabolic FLIM of CT26 tumors in vivo. (**A**) PLIM and FLIM images of tumors of different sizes. Tumor volume is indicated above the images. Scale bar = 50 μm. For FLIM: ex. 750 nm, reg. 450–490 nm. For PLIM: ex. 750 nm, reg. 570–640 nm. (**B**) Phosphorescence decay curves of the polymer probe PIr3 in the zones of normoxia (red line) and hypoxia (green line). Black or blue dots are the experimental data. (**C**) Scatter plot for cell measurements of PIr3 phosphorescence lifetime (τ PIr3) and NAD(P)H fluorescence lifetime (a_1_, %). The Pearson correlation r is shown. (**D**) Phosphorescence lifetime of the PIr3 polymer probe and the relative contribution of free NAD(P)H (a_1_, %) in the individual tumors numbered from 1 to 8. The median and the quartiles Q1 and Q3 are shown. Dots are the measurements in the cytoplasm of individual cells. (**E**) Comparisons of τ PIr3 and a_1_ NAD(P)H between tumors of different volumes. Box shows the median and the quartiles Q1 and Q3, whiskers show minimum and maximum. *n* = 8–10 fields of view from 2–4 tumors. (**F**) Dendrogram of hierarchical clustering showing the presence of two clusters in a cell population corresponding to the normoxic (green) and hypoxic (red) cells.

**Figure 10 ijms-23-10263-f010:**
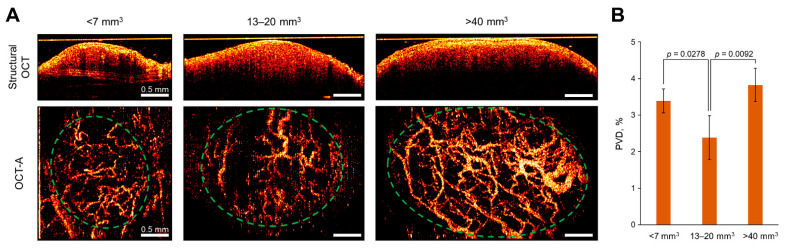
In vivo assessment of tumor microvascular network using OCT-A. (**A**) Representative structural OCT and OCT-A images of tumors. Scale bar = 0.5 mm. Tumors are shown by dashed circles. (**B**) Perfused vessel density in the tumors. Mean ± SD, *n* = 3–5. *p*-values are indicated on the diagram.

**Figure 11 ijms-23-10263-f011:**
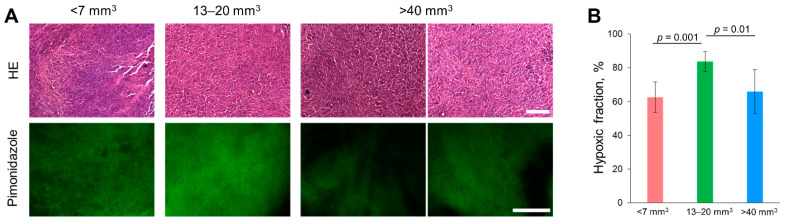
Histopathology and immunohistochemistry of CT26 tumors. (**A**) Upper row: representative histological slices of tumors, hematoxylin/eosin (HE) staining, initial magnification ×10. Scale bar = 270 μm. Bottom row: Immunohistochemical staining for hypoxia with pimonidazole. Scale bar = 20 μm. Initial magnification ×10. (**B**) Quantification of hypoxic fraction. Mean ± SD.

**Figure 12 ijms-23-10263-f012:**
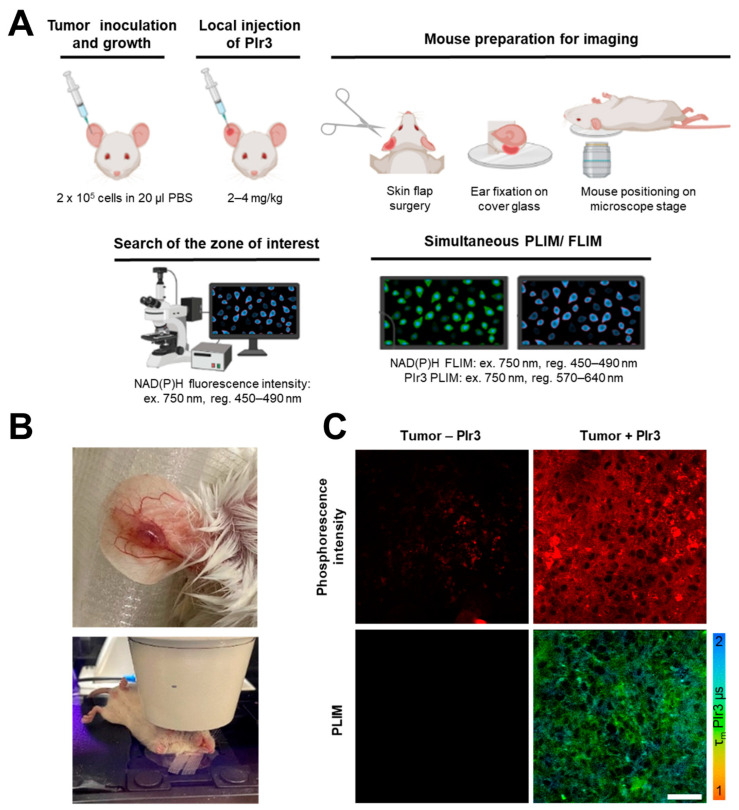
Intravital simultaneous PLIM of the polymer probe PIr3 and FLIM of NAD(P)H in a mouse tumor. (**A**) Mouse preparation for PLIM/FLIM microscopy. (**B**) Photographs of CT26 tumor on the ear on day 14 after implantation and the anesthetized mouse at the microscopy stage. (**C**) Phosphorescence intensity and lifetime images of the tumor without PIr3 probe (control) and with PIr3 probe injected locally. Scale bar = 50 µm.

**Table 1 ijms-23-10263-t001:** Photophysical characteristics of PIr1–PIr3 in CH_2_Cl_2_ solution.

Compound	λ_max_^abs^/nm (log ε) in CH_2_Cl_2_	λ_max_^em^/nm (in CH_2_Cl_2_)	Quantum Yield, % (in CH_2_Cl_2_)		Chromaticity Coordinates in the CIE Diagram (*x*; *y*)
			(a)	(b)	
PIr1	291 (5.12), 341 (4.97), 381 (4.76), 448 (4.54), 484 (4.39)	588, 629 sh	12.76	2.33	0.57; 0.41
PIr2	291 (5.32), 343 (5.14), 387 (4.93), 439 (4.76), 480 (4.62)	588, 629 sh	14.64	2.95	0.58; 0.41
PIr3	297 (5.03), 348 (4.86), 404 (4.08), 466 (4.63), 510 (4.53)	605	16.45	4.03	0.64; 0.36

(a) Degassed solution. (b) Aerated solution.

**Table 2 ijms-23-10263-t002:** Photophysical characteristics of PIr1–PIr3 in H_2_O solution.

Compound	λ_max_^abs^/nm (log ε) in H_2_O	λ_max_^em^/nm (in H_2_O)	Quantum Yield, % (in H_2_O)		Chromaticity Coordinates in the CIE Diagram (*x*; *y*)
			(a)	(b)	
PIr1	293 (4.74), 344 (4.57), 390 (4.31), 446 (4.11), 485 (3.94)	588, 629 sh	7.54	4.74	0.57; 0.40
PIr2	291 (4.75), 340 (4.61), 390 (4.34), 445 (4.16), 481 (3.99)	588, 629 sh	8.42	6.18	0.57; 0.39
PIr3	294 (4.62), 349 (4.39), 399 (4.12), 461 (3.94), 505 (3.74)	605	8.79	5.77	0.64; 0.36

(a) Degassed solution. (b) Aerated solution.

**Table 3 ijms-23-10263-t003:** Phosphorescence lifetimes of PIr1–PIr3 probes in degassed (τ_0_) and aerated (τ) aqueous solutions at room temperature. λ_ex_ = 375 nm.

	PIr1	PIr2	PIr3
τ_0_, μs	3.43 ± 0.07	3.15 ± 0.08	2.04 ± 0.03
τ, μs	2.13 ± 0.05	2.13 ± 0.05	1.41 ± 0.08
τ in PBS, μs	2.01 ± 0.09	1.98 ± 0.05	1.31 ± 0.05
τ in 5% BSA, μs	2.02 ± 0.07	1.95 ± 0.06	1.41 ± 0.05
τ in DMEM with 10% FBS, μs	1.91 ± 0.07	1.97 ± 0.07	1.33 ± 0.09
Dynamic range r = τ_0_/τ	1.61 ± 0.07	1.47 ± 0.07	1.44 ± 0.06
Photon counts (for τ)	5890	5600	13,823

## Data Availability

Not applicable.

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
