# Peer review of "Simultaneous Probing of Metabolism and Oxygenation of Tumors In Vivo Using FLIM of NAD(P)H and PLIM of a New Polymeric Ir(III) Oxygen Sensor"

_ijms, 2022, doi:10.3390/ijms231810263_

Round 1
Reviewer 1 Report
I think the article novelty was really exciting and oxygen sensor tool will be useful for cancer therapy.
The developing tumors are highly glycolytic, big tumors are in condition of mixed of hypoxia and normoxia. An increased hypoxic condition is an indication of metastasis in most solid cancers which promotes angiogenesis.
The novelty of this paper is that authors developed tool will help in measuring oxygen levels and glycolysis by indirect measuring of NADPH levels from in vivo tumors. This is a very useful tool in discovery field to find the efficacy of drug for better curing cancer therapy.
Reviewer 2 Report
In this research Authors applied the a new polymeric Ir(III)-based sensor PIr3 for monitoring oxygen in tumor cells in vivo. The applications research was proceeded by synthesis and characterization of three new polymeric probes PIr1-PIr3. Based on their cytotoxicity, the Authors selected PIr3 for applications as oxygen probe in tumor cells and mouse model. The obtained results clearly indicated that the new-synthesized oxygen sensor PIr3 is not only a promising tool for monitoring oxygen in vivo, but it gives opportunity for simultaneous probing of tumor cells metabolism using FLIM of NAD(P)H. Although the Authors discussed the limitation of their system, it is undeniably obvious that their work carries the valuable information to the other research groups focus on improving biotools for monitoring the development of cancer cells. The manuscript is also written well ( I noticed 1 or 2 typos) and with appropriate references and I recommend this paper to be published in International Journal of Molecular Sciences.
Reviewer 3 Report
A protocol of a dual FLIM/PLIM imaging in vivo is presented. These types of Fluorescence-based techniques are particularly valuable because they enable the measurement of different parameters within a single scan.
Hypoxic tumor cells produce a number of factors which can stimulate growth of new vasculature and promote invasiveness and metastatic spread. Moreover, hypoxic regions of tumors are frequently treatment resistant, altogether resulting in overall poor clinical outcome. Therefore, the development of improved probes for the measurement and imaging of tissue oxygen supply in vivo and in real time is of considerable interest for tumor diagnostics and characterization as well as for the evaluation of improved therapies.
This type of study could resolve inconsistencies between different studies on evaluation of tumor hypoxia at various stages of tumor growth or in tumors of different size, even for the tumors of one type.